# Investigation of Metrics for Assessing Human Response to Drone Noise

**DOI:** 10.3390/ijerph19063152

**Published:** 2022-03-08

**Authors:** Antonio J. Torija, Rory K. Nicholls

**Affiliations:** Acoustics Research Centre, University of Salford, Manchester M5 4WT, UK; r.k.nicholls@edu.salford.ac.uk

**Keywords:** drone noise, noise annoyance, noise metrics, loudness, sound quality metrics, subjective experiments

## Abstract

Novel electric air transportation is emerging as an industry that could help to improve the lives of people living in both metropolitan and rural areas through integration into infrastructure and services. However, as this new resource of accessibility increases in momentum, the need to investigate any potential adverse health impacts on the public becomes paramount. This paper details research investigating the effectiveness of available noise metrics and sound quality metrics (SQMs) for assessing perception of drone noise. A subjective experiment was undertaken to gather data on human response to a comprehensive set of drone sounds and to investigate the relationship between perceived annoyance, perceived loudness and perceived pitch and key psychoacoustic factors. Based on statistical analyses, subjective models were obtained for perceived annoyance, loudness and pitch of drone noise. These models provide understanding on key psychoacoustic features to consider in decision making in order to mitigate the impact of drone noise. For the drone sounds tested in this paper, the main contributors to perceived annoyance are perceived noise level (PNL) and sharpness; for perceived loudness are PNL and fluctuation strength; and for perceived pitch are sharpness, roughness and Aures tonality. Responses for the drone sounds tested were found to be highly sensitive to the distance between drone and receiver, measured in terms of height above ground level (HAGL). All these findings could inform the optimisation of drone operating conditions in order to mitigate community noise.

## 1. Introduction

A scenario with several drones (whether manned or unmanned) flying over cities and rural areas is now more likely than ever. To start with, there is a myriad of potential uses, from recreational to parcel delivery and even surveillance and law enforcement. There are substantial environmental and societal benefits associated with the wider expansion of the drone sector. For instance, medical deliveries to reduce waiting times [1] or reduction of carbon footprint in cargo transport and parcel delivery [2]. However, there are important concerns that can act as barriers for the wider adoption of these technologies: safety and privacy concerns, airspace management and visual and noise impact [3]. Although the main focus to date has been on the effects on human health, drone noise is also a source of concern for animal welfare [4].

The noise emission of drones and isolated drone propellers have been extensively studied [5,6,7,8,9,10,11,12,13,14]. Hui et al. [3] outlined the main sound generation mechanisms in drones, highlighting the contributions of rotor noise and electric motors. Schäffer et al. [15] conducted a systematic review (based on the PRISMA statement) on current methods for acoustic measurements and noise emission characteristics of drones; they also stated the requirements for a drone emission model and proposed a scheme for data acquisition. Overall, the sound emission of drones was found to be mainly influenced by drone size, configuration (number of rotors), payload, operating conditions and flight manoeuvres.

Schäffer et al. [15] carried out a systematic review on the effects of drone noise on humans, finding that the literature on the topic is still very limited. Christian and Cabell’s work [16] was pioneering in understanding the perception of drone noise and how it compared to other sources of transportation, such as road traffic. Gwak et al. [17] investigated the reported annoyance (in a subjective, laboratory-based experiment) to a series of drones of different size and under hover conditions. Hui et al. [3] investigated the perception (in a subjective, laboratory-based experiment) of a series of drone sounds, accounting for different operating conditions in terms of flight mode and height above the ground. Ivošević et al. [18] carried out a survey where a series of participants reported their in situ subjective assessment on drone noise during a series of drone operations in an open field.

Current evidence suggests that drone noise annoyance highly depends on loudness-related metrics [3,15,16]. However, Hui et al. [3] suggested that further research is needed to understand and quantify the effect of spectral and temporal factors (including tonality and impulsiveness) on drone noise annoyance. Schäffer et al. [15] also point out pure tones and high-frequency broadband noise as important contributors to drone noise annoyance. Tonal and high-frequency drone noise is also likely to increase its noticeability in existing soundscapes [19,20,21] and therefore lead to noise annoyance. None of these sound features are appropriately accounted for in current aircraft noise metrics, neither L_Aeq_-related nor even more sophisticated metrics, such as the effective perceived noise level (EPNL) [22]. Christian and Cabell [16] found that the A-weighted sound exposure level (L_AE_) was not able to account on its own for the difference in annoyance between drones and road vehicles. In their study, for a same value of L_AE_, drones were found to be more annoying than road vehicles. In fact, drones were reported to be as annoying as road vehicles with a 5.6 dB higher L_AE_. Torija et al. [21] also found that the L_Aeq_ metric does not account for the particular sound features related to drones, and that these particular features highly influence drone noise reported annoyance. Current regulation of drone noise is solely based on A-weighted sound power level (L_wA,max_), which, as suggested by Torija and Clark [23] might not provide an accurate picture of the noise impact of drones.

The aim of this paper is to investigate the perception of drone noise under controlled laboratory conditions in order to propose noise metrics for effectively assessing human response to drone noise. This research is framed within the perception-driven engineering approach, where sound quality metrics (SQMs) provide an accurate assessment of how the human auditory system responds to key sound features [23,24]. Gwak et al. [17] found that the noise annoyance of three drones (with maximum take-off mass (MTOM) ranging from 113.5 g to 11 kg) hovering was highly related to the SQMs loudness (perception of amplitude of the sound), sharpness (perception of high frequency) and fluctuation strength (perception of slow amplitude modulation). Torija and Li [25] found the noise annoyance of a series of drone (MTOM of 1.2 kg) flyovers highly related to a tonality metric and the interaction between loudness and sharpness. The research presented in this paper expands the number of drones under investigation to eight types, with payloads ranging from 1.2 to 11.8 kg and differing rotor number (including a contra-rotating configuration). Moreover, different manoeuvres are considered, including take-off, hover, flyover and landing, and varying heights above the ground are analysed. This comprehensive set of drone sounds aims to provide robust metric–noise-perception relationships for a representative sample of drone types. The main contributions of this paper are: (i) a better understanding of the noise perception of a comprehensive range of drone types and operating conditions; (ii) an assessment of the contribution of key acoustic and psychoacoustic features to drone noise perception in terms of perceived annoyance, perceived loudness and perceived pitch; and (iii) a quantification of changes in drone noise perception as a function of distance between drone and receiver. These findings can contribute to the development of noise metrics for assessing human response to drone noise and the definition of operational constrains in terms of distance to the receiver in order to minimise the community noise impact of drones.

The structure of the paper is as follows. Section 2 provides a brief overview of the main metrics for aircraft noise and SQMs. Section 3 describes both the selection of drone sounds used in this research and the methodology behind the subjective experiment investigating the relationship between drone noise and perception. Results are presented in Section 4 and include the relationship between drone noise perception and flyover altitude, effect of loudness on drone noise perception and metrics for drone noise assessment. Finally, results are discussed in Section 5, and conclusions and future work are stated in Section 6.

## 2. Overview of Noise Metrics

The impact of aircraft noise on communities is mainly assessed with exposure metrics related to A-weighted energy equivalent sound pressure level (L_Aeq_) [22,26]. Examples of these metrics are day–night level (DNL), day–evening–night level (DENL) and L_Aeq,16h_. At a vehicle level, broadband frequency-weighted sound pressure levels are used for aircraft noise assessment. These metrics include the maximum A-weighted sound level (L_A,max_) and the A-weighted sound exposure level (L_AE_). L_A,max_ is a common metric to assess sleep disturbance [27]. L_AE_ is numerically equivalent to the total sound energy of an aircraft overflight and therefore is very useful to compare total emission of different types of aircraft under different operating and payload conditions. However, these metrics do not account for other important features for noise annoyance, such as tonality [28,29]. The effective perceived noise level (EPNL) is the main metric for the noise certification of fixed-wing and rotary-wing aircraft [30] and is calculated according to a procedure developed by the Federal Aviation Administration [31]. This metric is based on the calculation of the perceived noise level (PNL) as proposed by Kryter [32]. The PNL is a descriptor for the overall perceived loudness, which is based on the Noy scale derived from a combination of amplitude and frequency [23]. The PNL is then corrected by an exposure duration factor and a tonality factor to obtain the EPNL metric. The tonality factor for the EPNL metric is based solely on the level of the strongest protruding tone, and therefore, as suggested by Torija et al. [22], might not be able to account for the perceptual effects of complex tonality (due to blade-passing frequency and harmonics of diverse rotors) typical in drones. Another limitation of the EPNL metric for assessing drone noise is that it does not consider frequency content above 10 kHz (which might be present in some drone types due to the operational frequency range of their electric motors) [7].

SQMs provide an accurate representation of huma hearing perception [15,33]. The most widely used SQMs are loudness (measured in sone), sharpness (measured in acum), fluctuation strength (measured in vacil), roughness (measured in asper), tonality (measured in tonality units—TU), and impulsiveness (measured in impulsiveness units—IU). Loudness measures the perception of sound intensity. Sharpness assesses the perception of spectral imbalances of a given sound towards the high-frequency region. Fluctuation strength and roughness assess the perception of slow and rapid fluctuations of the sound level. Tonality accounts for the perception of spectral irregularities of pure tones. Impulsiveness assesses the perception of sudden, abrupt increases in the sound level. All these metrics combined reflect the perception of sounds with various acoustic characteristics [15]. Further details on these SQMs can be found in Zwicker and Fastl [24] and Sottek et al. [34].

In addition to drone noise, as described in Section 1, SQMs have been used to develop noise-annoyance models for different aircraft types. For instance, Rizzi et al. [35] developed a model based on loudness, roughness and tonality to estimate noise annoyance for electric fixed-wing aircraft. Moreover, Krishnamurthy et al. [36] and Boucher et al. [37] found the SQMs sharpness, fluctuation strength and tonality as main contributors explaining noise annoyance for rotorcraft. In a study involving a psychoacoustic analysis of contra-rotating propellers, Torija et al. [33] suggested that loudness, tonality and fluctuation strength metrics are able to account for the perception of potential field interaction tones (dominating the sound emission with the rotor closely spaced). They also suggested that roughness and impulsiveness metrics are able to account for the perception of unsteadiness due to propeller–turbulence interaction noise.

## 3. Materials and Methods

### 3.1. Selection of Drone Sounds

A database of 44 drone sounds were carefully selected for the subjective experiment in order to assess perceived annoyance, loudness and pitch. The criteria for selection was to include sounds encompassing a wide variety of loudness level and other key psychoacoustic parameters for drone noise, including temporal and frequency characteristics (see [23]). The drone sounds were gathered from different sources, including in-house measurements [38], colleagues at the Volpe Center in the US [39] and colleagues from industry. Although the sounds were from various sources, they were selected to maintain a constant level of audio quality (as assessed by the authors of this paper). Each of the original sounds was edited to extract a sample of 4 sec long to use in the subjective experiment (to balance the gathering of perceptual data with participants’ fatigue [22]).

In total, sounds from eight types of drones were used, with the weight of these drones ranging from 1 to 12 kg. The chosen drone sounds also yielded a large variety of noise and operational characteristics. The drone operations included flyovers, hovering, manoeuvring, take-offs and landing. Furthermore, the drones were recorded performing these operations at differing heights above ground level (HAGL) of 2 to 60 m. The L_Aeq,4s_ of the drone sounds ranged between 37 dB and 71 dB. The full list of the 44 drone sounds with associated characteristics can be seen in Table 1. Differences in L_Aeq,4s_ between sounds from the same drone type with identical weight and HAGL might be attributed to small variations in operating and meteorological conditions, such as small differences in vehicle speed and different rotor rotational speeds to maintain vehicle stability under different wind profiles. It should also be noted that the DJI Phantom 3 was tested with varying payloads (see Torija et al. [38]).

As mentioned above, the drone sounds described in Table 1 were gathered from three different databases. Sounds S1 to S15 were recorded with a TASCAM DR-05 audio recorder, with sound pressure levels measured with a Norsonic 140 Class 1 sound level meter. These drone sounds were recorded in an open field in Alnmouth (northeast England). There were some other sounds present, including distant waves, birdsong and intermittent railway noise. Sounds S16 to S30 were recorded with a Brüel & Kjær 2250 Class 1 sound level meter with sound-recording capabilities. These drone sounds were recorded in an open field in Southampton. There were some other sounds present, including birdsong and a distant road. For further details see Torija et al. [38]. Sounds S31 to S44 were recorded by colleagues of the John A. Volpe National Transportation Systems Center in the Choctaw Nation of Oklahoma. Drone sounds were recorded using GRAS Model 40AO ½ inch pressure microphones and a Sound Devices 744T digital audio recorder. Sound pressure levels were measured with a Larson–Davis 831 Class 1 sound level meter. Recordings took place in a remote and quiet open field, with ambient sound mainly dominated by wind noise and some occasional aircraft flybys. For further details see Read et al. [39]. The ambient sound levels in all locations were considered sufficiently low so that they would not unduly influence the drone sound recordings. During the selection process, the databases available were carefully explored to discard any extraneous sounds.

Temporal and spectral characteristics of drone noise have been found to be highly influenced by the type of vehicle operation and meteorological conditions [15,23]. As the drone operates, adjustments in rotor operating conditions are made to maintain vehicle stability or to propel the vehicle in a given direction at a given speed (with a specific yaw, pitch and roll). These adjustments have been shown to significantly change the sound character [38], leading potentially to important changes in drone-noise perception. Furthermore, as described above, the operational HAGL was varied, as the drone noise spectrum contains higher frequencies than those of conventional aircraft. This high-frequency content is contributed to by harmonics of the rotor-blade-passing frequencies and the electric motors [5]. This, coupled with the decreased effects of air absorption due to the reduced operational HAGL (compared to conventional aircraft), increases the prominence of high-frequency content in drone noise and therefore should be considered in the subjective experiment.

### 3.2. Subjective Experiment

#### 3.2.1. Calibration of Test Stimuli

In order to set the sound pressure level of the drone sounds used in the subjective experiment to the L_Aeq,4s_ targets, a calibration process was designed. The control of the L_Aeq,4s_ of the test sounds is especially important when assessing the effect of operational factors, such as type of operation, vehicle weight and HAGL (and consequently loudness) on perception responses. The calibration setup included a class 1 BSWA308 sound level meter, a Bruel and Kjaer head and torso system (HATS), Norsonic front end type 336 microphone pre-amp, M-Audio M-Track 2X2M audio interface, AKG k 501 headphones, Audio Quest Dragonfly Red USB sound cards (24 bits 96 kHz) and a mainstream laptop for stimuli playback and recording. The calibration setup is presented in Figure 1.

To calibrate the drone-sound stimuli to the target L_Aeq,4s_ (shown in Table 1), a 1 kHz sine wave was played from the laptop through the headphones and recorded by the microphones in the HATS system. The level of the sine wave was measured using the sound level meter plugged in to the output of the microphone pre-amp. The difference between the measured level of the microphone pre-amp output and the original sine wave level (94 dB) was calculated to be the calibration factor to apply to the drone-sound stimuli. The calibration factor was applied to the drone-sound stimuli, and the stimuli were recorded through the HATS system so that SQM (and other noise metrics used)-analysis of the test sounds is representative of the stimuli as listened to by the participants. As the experiment was conducted online and not in a lab as originally planned (see Section 3.2.2), it was assumed that the sounds recorded through the calibration system described above were representative of the sounds heard by the participants (see Section 4.1 and Section 5 for further discussion).

#### 3.2.2. Experimental Procedure

Due to the COVID-19 pandemic and the restrictions put in place by the UK Government to mitigate its effects, a laboratory-based experiment was not possible to carry out. Instead, an online experiment was designed and built using the Web Audio Evaluation Toolkit (WAET) [40]. An interface was designed for the participants of the experiment in order to allow them to listen to the 44 drone-sound stimuli and provide their responses of perceived annoyance, loudness and pitch. The online experiment was designed to be completed in about 20 min (in order to maximise responses and completions of the full experiment).

The online experiment was accessible via personalised URL links in order to maintain anonymity and security between participant data. The online experiment was advertised on social media and to the staff and students at the University of Salford. Each person interested in participating was provided with a personalized URL link and a participant ID. Overall, 89 participants completed the online experiment in part, with 49 of them completing the full test (32 males and 17 females). Therefore, the responses of the 49 participants who completed the full test have been used for the analysis of this paper. The participants were instructed to complete the test in a quiet, distraction-free environment and to use high-quality headphones.

Each drone sound was presented individually to the participants. Once each drone sound was presented, the participants could listen to it as many times as required. Responses were then given using a set of sliders in the interface. Once the participants were satisfied with their responses, they could progress to the next stimulus, until the whole set of 44 test sounds were heard and assessed. The order of the stimuli for each participant was randomised.

Prior to the commencement of the experiment, each participant went through a pseudo-calibration stage in order to adjust the level of the UAV stimuli. Since the experiment was online and accessed remotely by the participants, the playback hardware used by each participant was unknown and highly likely to vary. This would lead to a variance in the playback quality and level of the stimuli between participants. To try to counter this, the participants were presented with the loudest and quietest UAV stimuli from the experiment and asked to adjust their playback volume so that the loudest stimulus was at a comfortable level and the quietest stimulus was still audible. Once the participant had appropriately adjusted their system playback level, they were asked to not adjust it for the remainder of the experiment. In addition to this, before starting the experiment, the participants were asked to match the sound levels of a series of tones in order to understand their frequency sensitivity (and also to detect substantial anomalies in the frequency response of the headphones used). A channel checking stage was also included to ensure that stereo playback was used. All 49 participants considered in this research paper appropriately passed these pseudo-calibration stages. In order to analyse the consistency of participants’ responses and the potential effect of using different sound reproduction settings (due to the experiment to be online), a statistical analysis was carried out (see Section 3.2.3 and Section 4.1 for more details).

The response variables considered were perceived annoyance, perceived loudness and perceived pitch. These response variables were chosen to be included in the subjective experiment as they relate to the amplitude of the sound event as well as various spectral and temporal characteristics of drone noise that have been shown to influence perception [15,16,21,23,25,33,38]. Perceived loudness was chosen as it is assumed to be a suitable response metric for explaining the effect of the distance of drone operation on perceived response. Perceived pitch was chosen as it is assumed to be a suitable response metric for explaining the effect of drone noise frequency content on perception. The questionnaire was designed according to the multi-dimensional scaling technique (MDS), which is based on dissimilarity ratings (see Susini et al. [41]). A continuous scale (from 0 to 1) was used for each subjective variable, labelled as follows: ‘Not Annoying’ at the left end and ‘Highly Annoying’ at the right end (perceived annoyance); ‘Not Loud’ at the left end and ‘Highly Loud’ at the right end (perceived loudness); and ‘Low Pitch’ at the left end and ‘High Pitch’ at the right end (perceived pitch).

The experiment reported in this paper is the first of two experiments carried out consecutively. The first experiment reported here, which focused on individual and isolated drone operations, investigated the influence of drone noise spectral and temporal characteristics on perception by analysing and assessing the relationships between participants’ responses and a selection of noise metrics (including SQMs). The second experiment (not reported here) will investigate the effects of drones on the soundscape in which they are operating.

#### 3.2.3. Data Analysis

To quantify the spectral and temporal characteristics of the drone-sound stimuli, the HEAD Acoustics ArtemiS Suite 12.5 software was used to calculate a series of SQMs (as described in Section 2). Loudness was calculated according to DIN45631/A1 [42]. This calculation method is based on Zwicker’s loudness model and includes a modification for time-varying signals. The calculation of sharpness was made according to the Aures method [43] due to the observably large variance in the loudness of the stimuli. Tonality was calculated according to the Aures/Terhardt tonality model [44]. Roughness, fluctuation strength and impulsiveness were calculated following the methods derived by Sottek [34,45]. For the calculation of the SQMs described above, the first 0.5 sec of each sound stimulus was omitted in order to remove any potential transient effect in the sound file that had resulted from editing the stimuli to 4 sec samples. As described by Torija et al. [33], the 5th percentile of the SQMs were used for the statistical analysis to investigate the perception of the drone noise samples tested in this paper. The PNL metric was calculated according to Kryter’s model [32] (see Section 2) with code developed in-house.

The statistical analyses were carried out with the IBM SPSS v.25 statistics software. A correlation analysis (including partial correlation) was implemented to give an initial insight into the relationship between each SQM and PNL and the perceived responses of annoyance, loudness and pitch for each drone sound stimulus. A multiple linear regression (MLR) analysis was also carried out to further determine the main contributors for the perceptual variables assessed in this paper. A forward stepwise-regression method (entry criterion for F-value ≤ 0.01) was implemented.

As the experiment was carried out online, with the participants reproducing the test sounds with different reproduction settings, two statistical tests were implemented to assess consistency between participant responses: (i) a Kendall’s W test to investigate the concordance in perceived annoyance, loudness and pitch for each drone sound between participants (see Section 4.1); and (ii) a multilevel analysis to identify the significance of subject-dependent responses and assess consistency in the perceived annoyance, loudness and pitch for the stimuli tested. The multilevel analysis was carried out according to Boucher et al. [37], with pooling of data between subjects. Pooling by subject creates a partial-pooling methodology and assumes normal distribution across subjects.

## 4. Results

### 4.1. Analysis of Consistency between Participant Responses

As discussed by Torija and Flindell [46], in experiments involving participants assessing sound stimuli it is expected that there would be a certain degree of variability in the participants’ responses. For this research, the experiment was completed online, with participants using different sound reproduction systems, and, therefore, a question on consistency of the responses might arise. To investigate inter-participant variability and agreement among the different participants in their responses of perceived annoyance, loudness and pitch for each of the test sounds, a statistical analysis was conducted. In the first step, it was found that the participants’ responses of perceived annoyance, loudness and pitch did not follow a normal distribution (based on both Kolmogorov–Smirnov and Shapiro–Wilk tests). A non-parametric, k-related sample statistic, Kendall’s W, was calculated for each perceptual variable, accounting for the whole set of 44 test sounds. Monte Carlo bootstrapping with 10,000 samples was implemented to ensure a robust calculation of the *p*-values. As can be seen in Table 2, very good agreement was found between participants in the responses of perceived annoyance and loudness for each of the test sounds (Kendall’s W ≥ 0.6, with W = 0 meaning no agreement among responses). Good agreement was also found for perceived pitch, although with a Kendall’s W value of about 0.4. As shown in Table 2, the *p*-values are smaller than 0.01, which allows certainty in rejecting the null hypothesis of no agreement between participants’ responses.

The coefficient of variation (CV) was also calculated as the standard deviation divided by the mean in order to check consistency in participants’ responses. As can be seen in Table 3, the CV for the three perceptual variables and for each test sound is consistently about 0.2–0.4 (which could be assumed to be an acceptable level). There are only a few test sounds where the CV increases notably. After further exploration, the CV of perceived annoyance and loudness reach higher values for test sounds S6, S12, S36 and S37, which all correspond to flyovers at high HAGL (see Table 1). For these sounds, with HAGL about 46 and 60 m, the reduction in loudness might make the contribution of other psychoacoustic factors more significant to the subjective responses, leading to less agreement in responses. The cases with the higher value of CV for perceived pitch correspond to sounds S41–S44, all from the same drone type (Gryphon GD28X). This particular drone model is based on a contra-rotating propulsion system. The spectral and temporal characteristics of this drone, with its overlapping propellers, seemed to lead to less agreement in participants’ responses of perceived pitch (although further research will be needed to fully understand the reasons behind this finding).

### 4.2. Peceived Annoyance, Loudness and Pitch as a Function of Height above Ground Level

The aggregated participant responses for each flyover drone sound were used to investigate the relationship between perceived annoyance, loudness and pitch with HAGL. As it can be seen in Figure 2 (perceived annoyance), Figure 3 (perceived loudness) and Figure 4 (perceived pitch), there is a strong logarithmic correlation between the three perceptual variables and HAGL. Although with some variability above 30 m, there is a clear trend of lower perceived values of annoyance and loudness (for the flyover sounds tested) as the HAGL increases. It is also important to note that the perceived loudness decays more rapidly with HAGL than the perceived annoyance, which might suggest the important contribution of psychoacoustic factors (and probably non-acoustic factors, such as perceived safety) other than loudness. Although there is a clear logarithmic correlation between perceived pitch and HAGL, the residual differences from the trendline are greater (see Figure 4). The perceived pitch might be expected to be associated with the high-frequency content of the test sound (accounted for by the sharpness metric). As the flyover is farther away, the high-frequency content becomes less prominent due to atmospheric absorption. However, a weaker logarithmic correlation with HAGL suggests that other psychoacoustic factors might have an important contribution to the perceived pitch.

In order to further investigate the relationship between different psychoacoustic factors and the perceived responses for the flyover drone sounds tested, a correlation analysis was carried out. Zero-order and partial correlation coefficients controlling for HAGL were calculated between PNL and the SQMs (loudness, sharpness, fluctuation strength, Aures tonality, roughness, impulsiveness) and perceived annoyance (Table 4), perceived loudness (Table 5) and perceived pitch (Table 6). Note that the partial-correlation analysis allowed measurement of the correlation coefficients whilst controlling the effect of HAGL.

As seen in Table 4 and Table 5, PNL, loudness and sharpness have the highest correlation (*p* ≤ 0.05) with perceived annoyance and loudness, respectively, both zero-order and when controlling for HAGL. This suggests that the different HAGL for the flyover drone sounds did not have any influence in the relationship between PNL, loudness and sharpness and responses of perceived annoyance and perceived loudness. For the case of perceived pitch (Table 6), the highest zero-order correlation is with PNL, loudness and sharpness, but when controlling for HAGL there is a statistically significant correlation with roughness as well. This might help explain the results shown in Figure 4, where it was suggested that other psychoacoustic factors were likely to explain the participants’ responses of perceived pitch as a function of HAGL.

### 4.3. Loudness and PNL vs. Perceived Responses

After investigating the correlation between PNL and the SQMs considered and the perceived responses for flyover drone sounds at varying HAGL, a bivariate correlation analysis was carried out with the whole set of test sounds. As seen in Table 7, PNL, loudness, sharpness and fluctuation strength have statistically significant correlation (*p* ≤ 0.05) with both perceived annoyance and perceived loudness. Responses on perceived annoyance and loudness seem to have a significant association to PNL, loudness and sharpness.

Table 7 also shows that PNL, loudness, sharpness, Aures tonality, roughness and impulsiveness have statistically significant correlation (*p* ≤ 0.05) with perceived pitch. For this particular case, sharpness has the highest correlation with perceived pitch, although contribution from the metrics listed above is observed as well.

An important finding, as shown in Table 7, is that PNL has a greater correlation with perceived annoyance and perceived loudness than the loudness SQM (and similar correlation to perceived pitch as loudness). The PNL metric was developed by Kryter [32] to assess the perception of jet aircraft noise. The frequency vs. sound pressure level (defined in the Noy scale) seems to be able to efficiently capture the perception of the amplitude and spectral characteristics of the drone sounds tested. The Noy scale, and therefore the PNL metric, assumes a significant sensitivity to higher frequency noise, which might explain its strong correlation with perceived annoyance and loudness. For this reason, it was decided to use the PNL and not the loudness metric for the subsequent regression analysis presented in Section 4.4.

### 4.4. Metrics for Drone Noise Assessment

An MLR analysis was undertaken to define the main contributors to perceived annoyance, perceived loudness and perceived pitch. All assumptions of MLR were checked before implementing the analysis:The linear relationship between predictors and dependent variables was verified in Table 7.The value of the variance inflation factor (VIF), well below 10, allowed the assumption of no multicollinearity in the data.The Durbin–Watson statistic was used to test the assumption of being independent. The value of this statistic was 1.30 (perceived annoyance), 1.52 (perceived loudness) and 1.39 (perceived pitch), allowing the assumption of the residuals being independent.Homoscedasticity was assumed by observing a random distribution of values in scatterplots between the regression-standardised predicted values and the regression-standardised residual values.Exploration of the P–P plots, with data points close to the observed vs. expected cumulative probability diagonal line, allowed the assumption of residuals being normally distributed.Values of Cook’s distance below 1 (max values of 0.12 for perceived annoyance, 0.83 for perceived loudness and 0.23 for perceived pitch) allowed the assumption of no influential cases biasing the MLR models presented.

As shown in Table 8, participants’ responses of perceived annoyance are mainly driven by PNL and sharpness. This suggests the perceived loudness and the high-frequency content of drone noise are key elements for perceived annoyance [17]. Perceived loudness is mainly driven by PNL and fluctuation strength. In this case, the beating effect due to the interaction between rotors seems to play an important role in perceived loudness [33]. The main contributors to perceived pitch are sharpness, roughness and Aures tonality. These results seem to indicate that pitch, as perceived by the participants, is highly influenced by the high-frequency and tonal content of drone noise, including the perceptual effect of the interaction between complex tones [22] typical of drone noise [23].

A multilevel analysis was also carried out to investigate whether the main contributors to perceived annoyance, loudness and pitch (as reported in Table 8) were consistent between participants. Table 9 shows the statistically significant predictors for perceived annoyance, loudness and pitch based on a multilevel analysis with subject-dependent intercepts and regression slopes. As seen in Table 9, there is consistency in the main contributors to the three perceptual variables between participants. The similarity of results in Table 8 and Table 9 suggest certainty in the predictors for the subjective models.

Following a ‘one-off’ approach previously implemented by Boucher et al. [37] and Torija et al. [21], the relative importance of a metric was assessed based on model accuracy (in terms of R^2^) when removing it from the analysis. Figure 5 confirms that participants’ responses of perceived annoyance and loudness are highly determined by PNL, while the responses of perceived pitch are highly determined by Aures tonality and especially roughness. These results suggest that rapid amplitude modulation, for instance due to unsteadiness of the sound signal, might affect the perception of pitch.

## 5. Discussion

The results presented in this paper, in terms of correlation between metrics and drone noise perception, are consistent with existing literature. In line with Hui et al. [3], the subjective responses evaluated in this paper correlate strongly with loudness-related metrics. Other SQMs driving responses of perceived annoyance were sharpness and fluctuation strength, similar to Gwak et al. [17]. Interestingly, the two metrics driving responses of perceived loudness were PNL and fluctuation strength. The contribution to fluctuation strength to perceived loudness (and also to perceived annoyance) is assumed to be due to accounting for the perceptual effect of interactions between rotors, however, further research is needed to validate this assumption. It has also been found that the perceived pitch for the drones assessed was mainly related to the high frequency, tonality and rapid amplitude modulation (described as due to interaction between discrete tones by Torija et al. [22]).

In Christian and Cabell [16], the annoyance response did not change significantly with drone altitude ranging from 10 to 100 m above ground level with other parameters held constant. However, this research has found the subjective responses evaluated changing significantly with HAGL. For instance, the perceived annoyance varied from about 0.5 to 0.4 when the HAGL changed from 20 to 60 m. Unlike Christian and Cabell [16], where only a SUI Endurance equipped with two-blade props was considered for analysis, this research investigated subjective responses vs. HAGL with flyovers of seven types of drones (DJI Inspire, Intel Falcon, DJI Matrice 200 and 600, DHI Phantom 3 and Yuneec Typhoon). This might be one of the reasons for the different findings, but further research is needed to investigate in more depth the relationship between drone noise perception and distance from the receiver. A better understanding of this relationship is key to defining operational constraints for drones in order to minimise community noise impact.

Due to the COVID-19 pandemic, only the subjective experiment was carried out, and not as it is usually done under controlled laboratory conditions. As recently reported by the technical committee of the Acoustical Society of America on Psychological and Physiological Acoustics, online experiments can provide access to larger sample sizes and ecologically valid responses, but at the cost of compromising the calibration process and finding inconsistencies in participant experiences [47]. Although the statistical tests reported in Section 4 confirmed consistency in participant responses, the results presented in this paper should be interpreted with caution, and important caveats should be considered:

(i) A careful process to calibrate the drone sounds to the target L_Aeq,4s_ (shown in Table 1) was carried out. Moreover, a pseudo-calibration stage was included in the online platform, where the participants adjusted their playback volume of the loudest stimulus to a comfortable level and the quietest stimulus to a just-audible level (see Section 3.2.2 for further details). The assumption was that even though the adjusted playback volume was different for each participant, the relative values of L_Aeq,4s_ of the individual stimuli were consistent (as they were calibrated in the laboratory as described in Section 3.1). However, the different playback hardware and quality and frequency response of each participant’s headphones might have altered the actual L_Aeq,4s_ of the individual stimuli as heard by the participants.

(ii) The participants were instructed to complete the test with adequate headphones and in a quiet environment. However, it can be assumed that the quality and frequency response of the headphones used and the background sound level where they completed the test varied between participants.

(iii) The online experiment was designed for the participants to reproduce the stimuli via headphones. There is uncertainty as to whether the headphones used by the participants were able to recreate the low frequency noise produced by the drones evaluated.

Therefore, the research findings presented and described in this paper will be validated in a subsequent experiment carried out under controlled laboratory conditions and reproducing the test sounds via a loudspeaker array.

## 6. Conclusions

This paper presents the results of a subjective experiment to investigate the noise perception of a comprehensive set of drone sounds encompassing different flying operations, size, weight and distance from the receiver. Based on a detailed statistical analysis, the responses of the participants for each drone sound were very consistent (even though the experiment was conducted online).

For the drone sounds tested, the participants’ responses of perceived annoyance were mainly driven by PNL and sharpness, confirming the significance of the high-frequency content present in drone noise. For perceived loudness, participants’ responses were mainly driven by PNL and fluctuation strength. In this case, the beating effect due to rotor interactions might affect the perception of loudness. Perceived pitch was found to be highly influenced by sharpness, Aures tonality and roughness. In this case, the perception of pitch seemed to be driven by the high-frequency and tonal content along with the rapid amplitude modulation due to the unsteadiness of the sound signal.

A robust logarithmic relationship was found between the perceived responses and distance from the receiver (quantified in terms of HAGL). An increase in HAGL of drone flyovers led to consistent reductions in perceived annoyance, perceived loudness and, to a lesser degree, perceived pitch. However, it was found that the perceived loudness declines more rapidly than perceived annoyance when HAGL increases. This means that increasing the distance of drone operations from communities is likely to lead to more substantial reductions in perceived loudness than perceived annoyance. This is likely due to other psychoacoustic factors, or even non-acoustic factors, on the perception of annoyance.

The findings of this research could facilitate better understanding of the key psychoacoustic factors to account for in order to mitigate community noise impact when planning drone operations. The metrics proposed in this paper could also aid the effective assessment of human response to drone noise.

Further research is needed to better understand the effects of drone noise on existing soundscapes, how ambient noise may mask drone noise, and the influence of this masking on the perception of drone operations.

## Figures and Tables

**Figure 1 ijerph-19-03152-f001:**
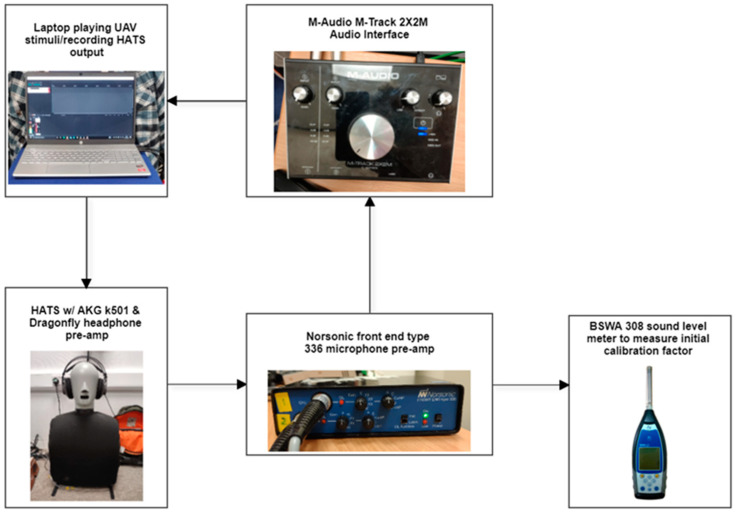
Calibration setup used for subjective test drone stimuli.

**Figure 2 ijerph-19-03152-f002:**
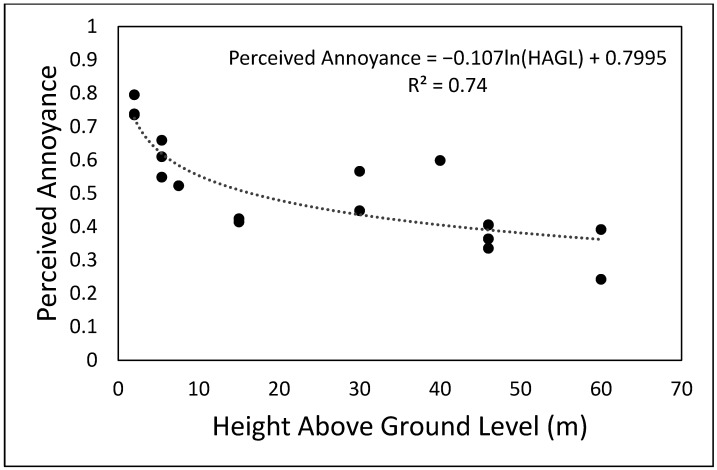
Perceived annoyance vs. height above ground level of the unmanned aerial vehicles under investigation during flyover operation.

**Figure 3 ijerph-19-03152-f003:**
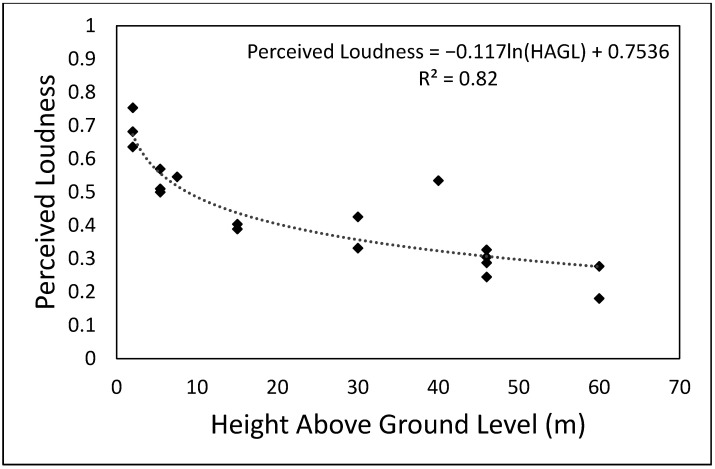
Perceived loudness vs. height above ground level of the unmanned aerial vehicles under investigation during flyover operation.

**Figure 4 ijerph-19-03152-f004:**
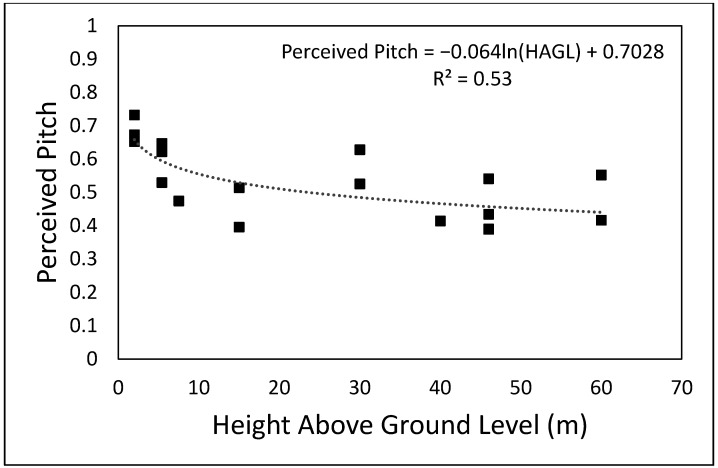
Perceived pitch vs. height above ground level of the unmanned aerial vehicles under investigation during flyover operation.

**Figure 5 ijerph-19-03152-f005:**
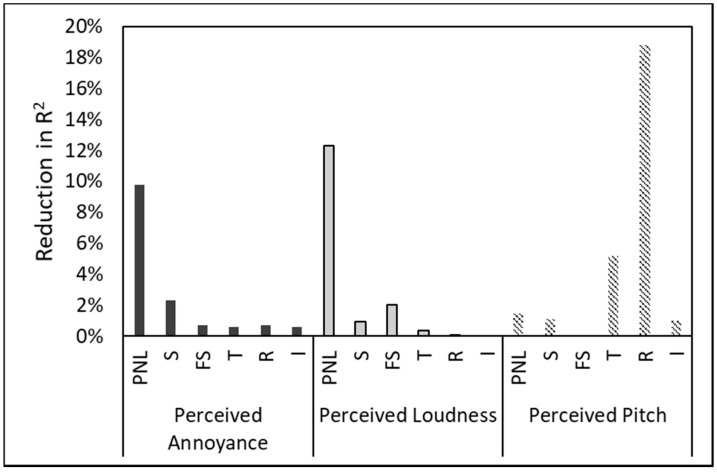
Reduction in R^2^ per predictor removed from multilevel model (using subject-dependent intercepts and regression slopes) for perceived annoyance, loudness and pitch.

**Table 1 ijerph-19-03152-t001:** Drone sounds used in the subjective experiment.

Sound ID	Drone Model	Drone Weight (kg)	Operating Procedure	Height above Ground Level, HAGL (m)	Calibrated L_Aeq,4s_
S1	DJI Inspire	2.85	Flyover	15	52
S2	DJI Inspire	2.85	Flyover	7.5	58
S3	DJI Inspire	2.85	Landing	7.5	64
S4	DJI Inspire	2.85	Takeoff	2	70
S5	Intel Falcon	1.2	Flyover	30	54
S6	Intel Falcon	1.2	Flyover	60	47
S7	DJI Matrice 600	9.1	Takeoff	3	71
S8	DJI Matrice 600	9.1	Hover	40	65
S9	DJI Matrice 600	9.1	Flyover	40	57
S10	DJI Mavic	0.743	Flyover	15	51
S11	DJI Mavic	0.743	Flyover	30	46
S12	DJI Mavic	0.743	Flyover	60	37
S13	DJI Mavic	0.743	Maneuvering	7.5	51
S14	DJI Mavic	0.743	Maneuvering	7.5	53
S15	DJI Mavic	0.743	Takeoff	7.5	59
S16	DJI Phantom 3	1.216	Maneuvering	2	68
S17	DJI Phantom 3	1.216	Takeoff	2	64
S18	DJI Phantom 3	1.216	Landing	2	62
S19	DJI Phantom 3	1.216	Hover	2	69
S20	DJI Phantom 3	1.216	Ascending	2	64
S21	DJI Phantom 3	1.216	Flyover	2	61
S22	DJI Phantom 3	1.216	Flyover	2	63
S23	DJI Phantom 3	1.216	Flyover	2	66
S24	DJI Phantom 3	1.216	Flyover	5.4	56
S25	DJI Phantom 3	1.216	Flyover	5.4	59
S26	DJI Phantom 3	1.216	Flyover	5.4	57
S27	DJI Phantom 3	1.216	Hover	2.2	62
S28	DJI Phantom 3	1.216	Hover	5.1	56
S29	DJI Phantom 3	1.216	Hover	2.2	67
S30	DJI Phantom 3	1.216	Hover	3.6	67
S31	DJI Matrice 200	4	Flyover	46	56
S32	DJI Matrice 200	4	Flyover	46	45
S33	DJI Matrice 200	4	Takeoff	30	50
S34	DJI Matrice 200	4	Landing	30	52
S35	DJI Matrice 200	4	Hover	1.2	56
S36	Yuneec Typhoon	2	Flyover	46	48
S37	Yuneec Typhoon	2	Flyover	46	44
S38	Yuneec Typhoon	2	Takeoff	30	46
S39	Yuneec Typhoon	2	Landing	30	52
S40	Yuneec Typhoon	2	Hover	1.2	57
S41	Gryphon GD28X	11.8	Takeoff	30	53
S42	Gryphon GD28X	11.8	Landing	30	54
S43	Gryphon GD28X	11.8	Maneuvering	30	57
S44	Gryphon GD28X	11.8	Hover	1.2	60

**Table 2 ijerph-19-03152-t002:** Results of the Kendall’s W statistic for the responses on perceived annoyance, loudness and pitch.

	Perceived Annoyance	Perceived Loudness	Perceived Pitch
Kendall’s W	0.60	0.64	0.41
*p*-value ^1^	0.00	0.00	0.00

^1^ *p*-value calculated with Monte Carlo bootstrapping with 10,000 samples.

**Table 3 ijerph-19-03152-t003:** Coefficient of variation for each test sound and for perceived annoyance, loudness and pitch.

Sound ID	Perceived Annoyance	Perceived Loudness	Perceived Pitch
S1	0.55	0.49	0.44
S2	0.32	0.31	0.39
S3	0.20	0.19	0.36
S4	0.17	0.18	0.36
S5	0.36	0.47	0.32
S6	0.62	0.67	0.41
S7	0.15	0.19	0.45
S8	0.36	0.35	0.47
S9	0.30	0.32	0.44
S10	0.48	0.43	0.35
S11	0.52	0.60	0.41
S12	0.87	0.83	0.58
S13	0.33	0.46	0.22
S14	0.28	0.35	0.27
S15	0.25	0.31	0.29
S16	0.12	0.17	0.25
S17	0.19	0.22	0.24
S18	0.24	0.25	0.27
S19	0.13	0.16	0.27
S20	0.15	0.20	0.22
S21	0.21	0.23	0.27
S22	0.23	0.23	0.28
S23	0.19	0.20	0.21
S24	0.35	0.38	0.41
S25	0.25	0.32	0.27
S26	0.31	0.35	0.28
S27	0.20	0.23	0.28
S28	0.34	0.39	0.36
S29	0.16	0.17	0.24
S30	0.22	0.18	0.25
S31	0.57	0.60	0.57
S32	0.54	0.60	0.58
S33	0.35	0.39	0.54
S34	0.36	0.41	0.46
S35	0.26	0.32	0.43
S36	0.64	0.74	0.44
S37	0.75	0.81	0.49
S38	0.49	0.55	0.31
S39	0.37	0.50	0.33
S40	0.58	0.72	0.51
S41	0.31	0.41	0.64
S42	0.38	0.34	0.64
S43	0.30	0.29	0.63
S44	0.24	0.29	0.73

**Table 4 ijerph-19-03152-t004:** Zero-order and partial correlation coefficients (controlling for height above ground level (HAGL)) between PNL and the SQMs (loudness, sharpness, fluctuation strength, Aures tonality, roughness and impulsiveness) and perceived annoyance. *p*-value shown in brackets.

	PNL	Loudness	Sharpness	Fluctuation Strength	Aures Tonality	Roughness	Impulsiveness
Zero-order	0.96 (*p* = 0.00)	0.91 (*p* = 0.00)	0.87 (*p* = 0.00)	0.24 (*p* = 0.33)	0.23 (*p* = 0.35)	0.08 (*p* = 0.75)	0.01 (*p* = 0.97)
Controlling for HAGL	0.88 (*p* = 0.00)	0.77 (*p* = 0.00)	0.76 (*p* = 0.00)	0.30 (*p* = 0.24)	−0.13 (*p* = 0.62)	−0.03 (*p* = 0.90)	−0.17 (*p* = 0.51)

**Table 5 ijerph-19-03152-t005:** Zero-order and partial correlation coefficients (controlling for height above ground level (HAGL)) between PNL and the SQMs (loudness, sharpness, fluctuation strength, Aures tonality, roughness and impulsiveness) and perceived loudness. *p*-value shown in brackets.

	PNL	Loudness	Sharpness	Fluctuation Strength	Aures Tonality	Roughness	Impulsiveness
Zero-order	0.98 (*p* = 0.00)	0.95 (*p* = 0.00)	0.85 (*p* = 0.00)	0.26 (*p* = 0.29)	0.32 (*p* = 0.20)	0.17 (*p* = 0.51)	0.01 (*p* = 0.97)
Controlling for HAGL	0.92 (*p* = 0.00)	0.86 (*p* = 0.00)	0.72 (*p* = 0.00)	0.38 (*p* = 0.14)	−0.03 (*p* = 0.91)	0.11 (*p* = 0.67)	−0.22 (*p* = 0.41)

**Table 6 ijerph-19-03152-t006:** Zero-order and partial correlation coefficients (controlling for height above ground level (HAGL)) between PNL and the SQMs (loudness, sharpness, fluctuation strength, Aures tonality, roughness and impulsiveness) and perceived pitch. *p*-value shown in brackets.

	PNL	Loudness	Sharpness	Fluctuation Strength	Aures Tonality	Roughness	Impulsiveness
Zero-order	0.68 (*p* = 0.00)	0.73 (*p* = 0.00)	0.76 (*p* = 0.00)	0.04 (*p* = 0.89)	0.09 (*p* = 0.73)	−0.35 (*p* = 0.16)	−0.08 (*p* = 0.76)
Controlling for HAGL	0.37 (*p* = 0.14)	0.49 (*p* = 0.05)	0.60 (*p* = 0.01)	−0.01 (*p* = 0.96)	−0.21 (*p* = 0.43)	−0.54 (*p* = 0.03)	−0.21 (*p* = 0.42)

**Table 7 ijerph-19-03152-t007:** Bivariate correlation coefficients (Pearson’s r coefficient) between PNL and the SQMs (loudness, sharpness, fluctuation strength, Aures tonality, roughness and impulsiveness) and perceived annoyance, loudness and pitch. *p*-value shown in brackets.

	PNL	Loudness	Sharpness	Fluctuation Strength	Aures Tonality	Roughness	Impulsiveness
Perceived Annoyance	0.96 (*p* = 0.00)	0.90 (*p* = 0.00)	0.90 (*p* = 0.00)	0.40 (*p* = 0.01)	0.25 (*p* = 0.10)	0.19 (*p* = 0.21)	−0.16 (*p* = 0.30)
Perceived Loudness	0.98 (*p* = 0.00)	0.92 (*p* = 0.00)	0.87 (*p* = 0.00)	0.47 (*p* = 0.00)	0.15 (*p* = 0.33)	0.29 (*p* = 0.06)	−0.08 (*p* = 0.59)
Perceived Pitch	0.47 (*p* = 0.00)	0.50 (*p* = 0.00)	0.55 (*p* = 0.00)	−0.00 (*p* = 0.98)	0.48 (*p* = 0.00)	−0.34 (*p* = 0.03)	−0.37 (*p* = 0.01)

**Table 8 ijerph-19-03152-t008:** Summary of multiple linear regression models to estimate perceived annoyance, loudness and pitch.

	R^2^	Adjusted R^2^	Predictors	Standardised Beta Coefficient	Variance Inflation Factor
Perceived Annoyance	0.93	0.93	PNL	0.72	4.15
			Sharpness	0.28	4.15
Perceived Loudness	0.97	0.97	PNL	0.95	1.18
			Fluctuation Strength	0.09	1.18
Perceived Pitch	0.61	0.59	Sharpness	0.56	1.13
			Roughness	−0.45	1.06
			Aures Tonality	0.32	1.08

**Table 9 ijerph-19-03152-t009:** Statistical significance (*p*-value) of predictors for perceived annoyance, loudness and pitch with subject-dependent intercepts and regression slopes.

Predictors	Perceived Annoyance	Perceived Loudness	Perceived Pitch
PNL	0.00	0.00	0.17
Sharpness	0.00	0.04	0.00
Fluctuation Strength	0.22	0.00	^1^
Aures Tonality	0.29	0.38	0.00
Roughness	0.21	0.82	0.00
Impulsiveness	0.14	^1^	0.50

^1^ Predictor redundant in multilevel analysis.

## Data Availability

Restrictions apply to the availability of the drone audio recordings used in this paper. Part of the audio recordings were obtained from John A. Volpe National Transportation Systems Center; and from Nathan Green. Access to this data can be requested to David R. Read, Christopher Cutler and Juliet Page (John A. Volpe National Transportation Systems Center); and Nathan Green (PhD student at the University of Salford’s Acoustics Research Centre). Part of the drone recordings were collected by A.J.T., and would be available upon request.

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
