# Peer review of "Investigation of Metrics for Assessing Human Response to Drone Noise"

_ijerph, 2022, doi:10.3390/ijerph19063152_

Round 1

Reviewer 1 Report

Metrics for Assessing Human Response to Drone Noise  

Antonio J. Torija and Rory K. Nicholls

This study is not very novel and most of the results are not particularly surprising (e.g. HAGL effects). However, it is a useful contribution that is worth publishing because it extends our knowledge of this area a little by examining the responses of people to sound produced by a substantial suite of drones varying in several characteristics that affect sound production performing several maneuvers.

I have only three minor comments that could be addressed in a revision:

  1. In outlining concerns about drones and drone noise, the possible effects on wildlife (especially birds) as well as humans should at least be mentioned. We share the planet with other species!
  2. It would have been useful to anonymously collect demographic data on the participants, particularly age, any known hearing deficiencies and home sound environment (e.g. rural vs. urban).
  3. In statistical analyses one should not use P value level (e.g. P<0.01) to infer how significant a result is and it would be better to give all P values exactly anyway. Although Kendall’s Coefficient of Concordance was used in analysis of some non-normally distributed variables, it is unclear whether other data were tested for conformity to statistical test prerequisites (normality, homoscedasticity etc.) and whether transformation was used if there was non-conformity. More generally, it is not great practice to analyse a data set repeatedly in multiple ways with different statistical analyses.

Author Response

Dear reviewer, please see the document attached.

Reviewer 2 Report

The presented article shows the considerable experience of the authors with writing similar texts. The first two chapters also provide a sufficient overview of the issue. The results are also relatively clear and properly explained, which is why I have only a few minor comments on the text.

On line 159: When LAeq is mentioned then there should be as a unit dB (without A). A-weighting is already shown in the name of the quantity.

Table 1: I don’t understand why for the drone DJI Phantom 3 there are so big differences between recordings at the same height (S21-S23, S24-S26 and S27 and S29), similarly for Yuneec Typhoon S36 and S37. There are differences up to 5 dB, which significantly affects the feeling of loudness and annoyance. I would expect some explanation.

Line 252: Why did the authors use the German standard instead of the ISO standard? In addition, the Moore-Glasberg model is more commonly used today for time varying sounds.

Figure 2: Personally, I would prefer to express the coefficient of variation in a table over a graph.

Lines 363-378 are typed in a smaller font.

I recommend checking the references. For some references the year is missing (5, 8), for others the year is given twice (15, 31), for some the source (journal or proceedings) is not obvious (25, 33, 35, 39).

Author Response

(The authors gave the same response as above.)

Reviewer 3 Report

Thanks for your submission. It's a very timely an interesting article given the growth in the UAV UAS market. Overall, the study seems solid and the methodology and results are substantiated. However, it seems like Gwak published a paper in 2021 dealing with psychoacoustics and drones (https://www.ncbi.nlm.nih.gov/pmc/articles/PMC8430946/ Quantification of the Psychoacoustic Effect of Noise from Small Unmanned Aerial Vehicles). this paper isn't cited and I think it might be worth seeing if can reframe the paper to better situate your findings. Your findings have value still, but this can provide a better context for your work.

A few comments

  • I think the title can be clearer. The title almost implies that you have developed new metrics but in fact, you are repurposing existing metrics for drones. I would suggest updating the title to something more specific and less general.
  • The background section is thorough, but i think the Gwak paper earlier needs to be included--and it might affect the structure of the paper 
  • I'd be more clear at the end of the introduction or at the end of the background about what your contributions are. I'd be more explicit. Lines 82 until the end of the section show some of it, but tell us what your contribution is and why it's significant. 
  • With your methodology, the challenge of online is quite difficult for an experiment like this. You show consistency with the statistics, but I think there should be some discussion about the potential vast difference in frequency response of an online test. I would be skeptical that these headphones that consumers wear can adequately recreate the low frequency response of drones, much less the rumbling that one experiences with drones. How do you account for this in your experiment? Consistency is one thing--everyone may have headphones that don't have sufficient bass response--so consistency does not surprise me. How can you validate your results for the lack of bass and rumbling that typical headphones do not recreate?
  • Figure 2, I'm not sure a line plot really accurately describes the graph--considering these are discrete test sounds, the way the lines are interpolated between test sounds, makes it appear to be a continuous graph when it is not. I would suggest a more clear way of depicting that data.
  • Table 8, please label what those numbers depict. It is a p value? 
  • Line 31-32--this is not a complete sentence.

Overall, besides the title, headphone freq response question, and the Gawk paper, this is a solid paper. You took care of the data, the statistics, and the findings are appropriate. I think some revision to update these things and contextualize your results with these aspects would make this a solid paper. 

Author Response

Dear Reviewer, please see the document attached.

Reviewer 4 Report

Section 3 must be improved. Describe in detail the equipment used to make the measurements of the noise emitted from drone e for recording the sound emitted from drones. Extract this data from the datasheet of the instrumentation manufacturer. To make reading the specifications of the instruments more immediate, you can insert them in a table, listing the instruments used and the specific characteristics for each. The recordings of the noise emitted by the drones were made outdoors. In this case not only the weather conditions affect the recordings but also the background noise. Authors must describe in detail the urban context in which these recordings took place. They must describe in detail any other sources of noise that obviously affect the perception of the noise. The authors carried out a remote listening test, it is necessary to clarify how it was possible to ensure that all participants reproduced the sounds in the same conditions. The authors must describe in detail how the listening tests and the questionnaire for the assessment of psychological aspects were prepared. These are the essential elements of the work and are not adequately dealt with. Authors must specify the standards used to prepare these tests (for example ISO-15666). How were the subjects prepared? Was a hearing test performed? Or have questions been asked about this? All this information must be detailed.

Section 4 must be improved. Improve the quality of figures. In Figures 3 and 4 the upper border is missing. A detailed discussion of the results obtained is missing. Try to summarize what was obtained and try to extract useful information from the work carried out.

38) Do not use abbreviation such as i.e. I have seen that you often use this abbreviation, so I will not repeat this advice again, it also applies to the other occurrences.

42) Do not use abbreviation such as e.g.) I have seen that you often use this abbreviation, so I will not repeat this advice again, it also applies to the other occurrences.

53) Add more references to works that have already dealt with the topic, for example,” Methods for attenuation of unmanned aerial vehicle noise”,” Acoustical unmanned aerial vehicle detection in indoor scenarios using logistic regression model”, :” Research for the presence of unmanned aerial vehicle inside closed environments with acoustic measurements”

120-123) Add references to allow readers to learn more about the following topics: Loudness, Sharpness, Fluctuation Strength, Roughness, Tonality, and Impulsiveness.

144-153) How the drone sound was recorded? What recorder has been used? Add this information.

154-160) Add detailed information about instrumentation used to obtain data showed in table 1 (LAeq,4s, Height Above Ground Level).

164-176) The recordings of the noise emitted by the drones were made outdoors. In this case not only the weather conditions affect the recordings but also the background noise. Authors must describe in detail the urban context in which these recordings took place. They must describe in detail any other sources of noise that obviously affect the perception of the noise.

190-202) Since the sounds will be reproduced by the participants' instruments, it is not clear what the calibration procedure was for. Authors need to clarify.

207-208) Since the authors used this interface, it would have been useful to play the sounds in this interface and then set the volumes of the machines used for playback.

212-213) How were the participants selected? No information is shown on the sample of participants and whether it is representative of the population. This information needs to be added.

222-232) How did the participants listen to the sounds? With headphones? If so, it would have been advisable to collect the characteristics of the headphones used. If it was administered without headphones, what do we know about background noise?

363-378) Check the format of the text, the dimension of the front appears reduced.

380) Make the table 3 fit on the same page.

Author Response

Section 3 must be improved. Describe in detail the equipment used to make the measurements of the noise emitted from drone e for recording the sound emitted from drones. Extract this data from the datasheet of the instrumentation manufacturer. To make reading the specifications of the instruments more immediate, you can insert them in a table, listing the instruments used and the specific characteristics for each. The recordings of the noise emitted by the drones were made outdoors. In this case not only the weather conditions affect the recordings but also the background noise. Authors must describe in detail the urban context in which these recordings took place. They must describe in detail any other sources of noise that obviously affect the perception of the noise. The authors carried out a remote listening test, it is necessary to clarify how it was possible to ensure that all participants reproduced the sounds in the same conditions. The authors must describe in detail how the listening tests and the questionnaire for the assessment of psychological aspects were prepared. These are the essential elements of the work and are not adequately dealt with. Authors must specify the standards used to prepare these tests (for example ISO-15666). How were the subjects prepared? Was a hearing test performed? Or have questions been asked about this? All this information must be detailed.

Authors’ response:  Dear reviewer, on behalf of the authors, I would like to thank you for your comments and suggestions which have certainly helped us to improve the quality of the manuscript.

The authors fully agree with the statement on drone sound recordings.  Further details have now been provided in Section 3.1:

‘The drone sounds described in Table 1 were gathered from three different databases.  Sounds S1 to S15 were recorded with a TASCAM DR-05 audio recorder, with sound pressure levels measured with a Norsonic 140 Class 1 sound level meter.  These drone sounds were recorded in an open field in Alnmouth (North East of England).  There were some sounds present, including distant waves, birdsong and intermittent railway noise.  Sounds S16 to S30 were recorded with a B&K 2250 Class 1 sound level meter with sound recording capabilities.  These drone sounds were recorded in an open field in Southampton.  There were some sounds present, including birdsong and a distant road.  For further details see Torija et al. [39].  Sounds S31 to S44 were recorded by colleagues of the John A. Volpe National Transportation Systems Center in the Choctaw Nation of Oklahoma.  Drone sounds were recorded using GRAS Model 40AO ½ inch pressure micro-phones and a Sound Devices 744T digital audio recorder.  Sound pressure levels were measured with a Larson-Davis 831 Class 1 sound level meter.  Recordings took place in a remote and quiet open field, with ambient sound mainly dominated by wind noise and some occasional aircraft flybys. For further details see Read et al. [40] The ambient sound levels in all locations were considered sufficiently low so that they would not unduly in-fluence the drone sound recordings.  During the selection process, the databases available were carefully explored to discard any extraneous sounds.’

The authors agree with your concerns regarding the likely listening experiences among participants. A comprehensive statement regarding the limitations of the experiment and the research findings has now been included at the end of the new Discussion section.

‘Due to the COVID-19 pandemic, the subjective experiment was carried out only, and not as usually done under controlled laboratory conditions.  As recently reported by the technical committee of the Acoustical Society of America on Psychological and Physio-logical Acoustics, online experiments can provided access to larger sample sizes and eco-logically valid responses, but at the cost of compromising the calibration process and finding inconsistencies in participant experiences [47].  Although the statistical tests re-ported in Section 4 confirmed consistency in participants responses the results presented in this paper should be interpreted with caution, and important caveats should be considered:

(i) A careful process to calibrate the drone sounds to the target LAeq,4s (shown in Table 1) was carried out.  Moreover, a pseudo-calibration stage was included in the online platform where the participants adjusted their playback volume of the loudest stimulus to a comfortable level and the quietest stimulus to a just audible level (see Section 3.2.2 for further details).  The assumption was that even though the adjusted playback volume was different for each participant, the relative values of LAeq,4s of the individual stimulus were consistent (as they were calibrated in the laboratory as described in Section 3.1). However, the different playback hardware, and quality and frequency response of the participant’s headphone might have altered the actual LAeq,4s of the individual stimulus as heard by the participants.

(ii) The participants were instructed to complete the test with adequate headphones, and in a quiet environment.  However, it can be assumed that the quality and frequency response of the headphones used and the background sound level where they completed the test varied between participants.

(iii) The online experiment was designed for the participants to reproduce the stimuli via headphones.  There is uncertainty as to whether the headphones used by the participants were able to recreate the low frequency noise produced by the drones evaluated.

Therefore, the research findings presented and described in this paper will be vali-dated in a subsequent experiment carried out under controlled laboratory conditions and reproducing the test sounds via a loudspeaker array.’

Furthermore, more details have been provided regarding the experimental procedure of the listening experiment.  See Section 3.2.2:

‘The online experiment was accessible via personalised URL links to maintain anonymity and security between participant data. The online experiment was advertised on social media, and within the staff and students at the University of Salford.  Each person interested in participating was provided with a personalized URL link and a participant ID.  Overall, 89 participants completed the online experiment in part, with 49 of them completing the full test (32 males and 17 females).  Therefore, the responses of the 49 participants completing the full test have been used for the analysis of this paper. The participants were instructed to complete the test in a quite environment, in order to avoid distractions, and reproducing the drone sounds via high quality headphones.

Each drone sound stimulus was presented individually to the participants.  Once each drone sound was presented, the participants could listen to it as many times as required. Responses were then given using a set of sliders in the interface. Once the participants were satisfied with their responses, they could progress to the next stimulus, until the whole set of 44 test sounds were heard and assessed. The order of the stimuli for each participant was randomised.

Prior to the commencement of the subjective experiment, each participant went through a pseudo-calibration stage, in order to adjust the level of the UAV stimuli. Since the experiment was online and accessed remotely by the participants, the playback hardware used by each participant unknown and is highly likely to vary. This would lead to a variance in the playback quality and level of the stimuli between participants. To try and counter this, the participants were presented with the loudest and quietest UAV stimuli from the experiment and asked to adjust their playback volume so that the loudest stimulus was at a comfortable level, and the quietest stimulus was still audible.  Once the participant had appropriately adjusted their system playback level, they were asked to not adjust if for the remainder of the experiment. In addition to this, before starting the experiment, the participants were asked to match the sound levels of a series of tones in order to understand their frequency sensitivity (and also to detect substantial anomalies in the frequency response of the headphones used).  A channel checking stage was also included to ensure that stereo playback was used. All 49 participants considered in this research paper appropriately passed these pseudo-calibration stages.  In order to analyse the consistency of participants’ responses, and the potential effect of using different sound reproduction settings (due to the experiment to be online), a statistical analysis was carried out (see section 3.2.3 and 4.1 for more details).

The response variables considered were perceived annoyance, perceived loudness and perceived pitch.  These response variables were chosen to be included in the subjective experiment as they relate to the amplitude of the sound event as well as various spectral and temporal characteristics of drone noise that have been shown to influence perception [15, 16, 21, 23, 25, 33, 38].  Perceived loudness was chosen as it is assumed to be a suitable response metric for explaining the effect of the distance of drone operation on perceived response. Perceived pitch was chosen as it is assumed to be a suitable response metric for explaining the effect of drone noise frequency content on perception. The questionnaire was designed according to the Multi-Dimensional Scaling technique (MDS), which is based on dissimilarity ratings (see Susini et al. [41]). A continuous scale (from 0 to 1) was used for each subjective variable, labelled as follows: ‘Not Annoying’ at the left end and ‘Highly Annoying’ at the right end (perceived annoyance); ‘Not Loud’ at the left end and ‘Highly Loud’ at the right end (perceived loudness); and ‘Low Pitch’ at the left end and ‘High Pitch’ at the right end (perceived annoyance).’

Section 4 must be improved. Improve the quality of figures. In Figures 3 and 4 the upper border is missing. A detailed discussion of the results obtained is missing. Try to summarize what was obtained and try to extract useful information from the work carried out.

Authors’ response: As requested by the reviewer, the quality of the figures has been improved.  Moreover, a Discussion section has been added (see new Section 5) to summarise the research findings and contextualise them in the current state-of-the-art.

38) Do not use abbreviation such as i.e. I have seen that you often use this abbreviation, so I will not repeat this advice again, it also applies to the other occurrences.

Authors’ response: Dear reviewer, the authors have carefully edited the manuscript as requested.

42) Do not use abbreviation such as e.g.) I have seen that you often use this abbreviation, so I will not repeat this advice again, it also applies to the other occurrences.

Authors’ response: Dear reviewer, the authors have carefully edited the manuscript as requested.

53) Add more references to works that have already dealt with the topic, for example,” Methods for attenuation of unmanned aerial vehicle noise”,” Acoustical unmanned aerial vehicle detection in indoor scenarios using logistic regression model”, :” Research for the presence of unmanned aerial vehicle inside closed environments with acoustic measurements”

Authors’ response: Many thanks for the recommended references.  These have now been added to the manuscript.

120-123) Add references to allow readers to learn more about the following topics: Loudness, Sharpness, Fluctuation Strength, Roughness, Tonality, and Impulsiveness.

Authors’ response: As recommended specific references have been suggested for further details on these Sound Quality Metrics.  See Section 2:

‘Further details on these SQMs can be found in Zwicker and Fastl [25] and Sottek et al. [35].’

144-153) How the drone sound was recorded? What recorder has been used? Add this information.

Authors’ response: Dear reviewer, details about drone sound recordings and instrumentation used have been added to Section 3.1.

154-160) Add detailed information about instrumentation used to obtain data showed in table 1 (LAeq,4s, Height Above Ground Level).

Authors’ response: Dear reviewer, details about drone sound recordings and instrumentation used have been added to Section 3.1.

164-176) The recordings of the noise emitted by the drones were made outdoors. In this case not only the weather conditions affect the recordings but also the background noise. Authors must describe in detail the urban context in which these recordings took place. They must describe in detail any other sources of noise that obviously affect the perception of the noise.

Authors’ response: Dear reviewer, details about drone sound recordings and instrumentation used have been added to Section 3.1.

190-202) Since the sounds will be reproduced by the participants' instruments, it is not clear what the calibration procedure was for. Authors need to clarify.

Authors’ response: Further details about the calibration process have been provided in Section 3.2.2.  This issue has also been discussed as a limitation of the study (see Discussion section).

207-208) Since the authors used this interface, it would have been useful to play the sounds in this interface and then set the volumes of the machines used for playback.

Authors’ response: It should be noted that the experiment was carried out fully online, and the experimented did not have access to the participants machines.  This option was explored, but after careful consideration it was discarded as this would be against the restrictions put in place by the UK Government to mitigate the effects of COVID-19 pandemic.

212-213) How were the participants selected? No information is shown on the sample of participants and whether it is representative of the population. This information needs to be added.

Authors’ response: This has now been clarified in the manuscript (Section 3.2.2).

‘The online experiment was advertised on social media, and within the staff and students at the University of Salford.  Each person interested in participating was provided with a personalized URL link and a participant ID.  Overall, 89 participants completed the online experiment in part, with 49 of them completing the full test (32 males and 17 females).  Therefore, the responses of the 49 participants completing the full test have been used for the analysis of this paper.’

222-232) How did the participants listen to the sounds? With headphones? If so, it would have been advisable to collect the characteristics of the headphones used. If it was administered without headphones, what do we know about background noise?

Authors’ response: This has now been clarified in the manuscript (Section 3.2.2).

‘The participants were instructed to complete the test in a quiet environment, in order to avoid distractions, and reproducing the drone sounds via high quality headphones.’

363-378) Check the format of the text, the dimension of the front appears reduced.

Authors’ response: This has been amended as requested.

380) Make the table 3 fit on the same page.

Authors’ response: This has been amended as requested.

Round 2

Reviewer 4 Report

The authors addressed all the reviewer's comments with sufficient attention and modified the paper consistently with the suggestions provided. The new version of the paper has improved significantly both in the presentation that is now much more accessible even by a reader not expert in the sector, and in the contents that now appear much more incisive. The detailed description of the figures makes them easier to understand for the reader.

Minor revision:

186)Table 1 goes to two pages, or you make it appear on a single page or you have to repeat the header.

389)Table 3 goes to two pages, or you make it appear on a single page or you have to repeat the header.

394) You seem to have deleted the whole caption

417)Figure 2 appears to be duplicated.

423) Figure 3 appears to be duplicated.

430)Figure 4 appears to be duplicated.

554) Figure 5 appears to be duplicated.

Author Response

Dear Reviewer,

please see the document attached for answers to your queries.

Kind regards,

Dr Antonio J Torija Martinez
